# In Vivo Inhibition of TRPC6 by SH045 Attenuates Renal Fibrosis in a New Zealand Obese (NZO) Mouse Model of Metabolic Syndrome

**DOI:** 10.3390/ijms23126870

**Published:** 2022-06-20

**Authors:** Zhihuang Zheng, Yao Xu, Ute Krügel, Michael Schaefer, Tilman Grune, Bernd Nürnberg, May-Britt Köhler, Maik Gollasch, Dmitry Tsvetkov, Lajos Markó

**Affiliations:** 1Department of Nephrology/Intensive Care, Charité—Universitätsmedizin Berlin, Corporate Member of Freie Universität Berlin and Humboldt-Universität zu Berlin, 10117 Berlin, Germany; zhihuang.zheng@charite.de; 2Experimental and Clinical Research Center, a Joint Cooperation of the Charité—University Medicine Berlin and Max Delbrück Center for Molecular Medicine in the Helmholtz Association, 13125 Berlin, Germany; may-britt.koehler@charite.de; 3Department of Internal Medicine and Geriatrics, University Medicine Greifswald, 17475 Greifswald, Germany; yao.xu@med.uni-greifswald.de; 4Rudolf Boehm Institute for Pharmacology and Toxicology, Leipzig University, 04107 Leipzig, Germany; ute.kruegel@medizin.uni-leipzig.de (U.K.); michael.schaefer@medizin.uni-leipzig.de (M.S.); 5Department of Molecular Toxicology, German Institute of Human Nutrition Potsdam-Rehbruecke (DIfE), 14558 Nuthetal, Germany; tilman.grune@dife.de; 6DZHK (German Centre for Cardiovascular Research), Partner Site, 10785 Berlin, Germany; 7Department of Pharmacology, Experimental Therapy and Toxicology and Interfaculty Center of Pharmacogenomics and Drug Research, University of Tübingen, 72076 Tübingen, Germany; bernd.nuernberg@uni-tuebingen.de; 8Berlin Institute of Health at Charité-Universitätsmedizin Berlin, 10178 Berlin, Germany; 9Charité-Universitätsmedizin Berlin, Corporate Member of Freie Universität Berlin and Humboldt-Universität zu Berlin, 10117 Berlin, Germany

**Keywords:** TRPC6, UUO, NZO mice, inflammation, fibrosis, CKD, SH045

## Abstract

Metabolic syndrome is a significant worldwide public health challenge and is inextricably linked to adverse renal and cardiovascular outcomes. The inhibition of the transient receptor potential cation channel subfamily C member 6 (TRPC6) has been found to ameliorate renal outcomes in the unilateral ureteral obstruction (UUO) of accelerated renal fibrosis. Therefore, the pharmacological inhibition of TPRC6 could be a promising therapeutic intervention in the progressive tubulo-interstitial fibrosis in hypertension and metabolic syndrome. In the present study, we hypothesized that the novel selective TRPC6 inhibitor SH045 (larixyl N-methylcarbamate) ameliorates UUO-accelerated renal fibrosis in a New Zealand obese (NZO) mouse model, which is a polygenic model of metabolic syndrome. The in vivo inhibition of TRPC6 by SH045 markedly decreased the mRNA expression of pro-fibrotic markers (*Col1α1*, *Col3α1*, *Col4α1*, *Acta2*, *Ccn2*, *Fn1*) and chemokines (*Cxcl1*, *Ccl5*, *Ccr2*) in UUO kidneys of NZO mice compared to kidneys of vehicle-treated animals. Renal expressions of intercellular adhesion molecule 1 (ICAM-1) and α-smooth muscle actin (α-SMA) were diminished in SH045- versus vehicle-treated UUO mice. Furthermore, renal inflammatory cell infiltration (F4/80+ and CD4+) and tubulointerstitial fibrosis (Sirius red and fibronectin staining) were ameliorated in SH045-treated NZO mice. We conclude that the pharmacological inhibition of TRPC6 might be a promising antifibrotic therapeutic method to treat progressive tubulo-interstitial fibrosis in hypertension and metabolic syndrome.

## 1. Introduction

Chronic kidney disease (CKD) is characterized by progressive loss of kidney function. The main risk factors of developing CKD are the combination of obesity, diabetes and hypertension, which is commonly referred to as metabolic syndrome. Other contributors are autoimmune diseases (e.g., glomerulonephritis), environmental exposures and genetic risk factors [1,2]. Morphologically, persistent low-grade renal inflammation and tubulointerstitial fibrosis are key hallmarks of CKD [3,4]. The complex interplay of fibroblasts, lymphocytes, tubular, and other cell types in the kidney lead to excessive extracellular matrix deposition and the further deterioration of renal function [5,6]. Although unspecific treatments strategies are available (e.g., medications lowering blood pressure), CKD progression is still poorly controlled.

In recent years, novel drug targets, such as transient receptor potential cation channel, subfamily C, and member 6 (TRPC6), emerged [7,8]. *TRPC6* mutations lead to glomerular injury and proteinuria, presumably involving the Ca^2+^ signaling pathway and resulting in progressive kidney failure [9,10,11,12]. Both TRPC6 gain-of-function and loss-of-function cause familial forms of focal segmental glomerulosclerosis (FSGS) [11,13]. Interestingly, in a murine model of kidney injury (unilateral ureteral obstruction (UUO)), *Trpc6*^−/−^ deficiency and pharmacological blockade with BI-749327 ameliorated renal fibrosis in C57BL/6J mice [7,8]. Remarkably, these beneficial effects were not observed in the acute stage of kidney injury (AKI) [14]. Thus, TRPC6 inhibition may have effects on renal fibrogenesis during AKI-to-CKD transition. Given this state of affairs, TRPC6 inhibition seems to represent a promising new therapeutic approach to combat progressive renal failure since it potentially affects CKD at later stages after kidney injury. However, it is unknown whether TRPC6 inhibition is effective for inhibiting progressive tubulo-interstitial fibrosis in hypertension and metabolic syndrome.

Recently, by the chemical diversification of (+)-larixol originating from *Larix* decidua resin traditionally used for inhalation, its methylcarbamate congener, named SH045, was developed as a novel, highly potent, subtype-selective inhibitor of TRPC6 [15]. In the present study, we hypothesized that this novel selective TRPC6 inhibitor (SH045) [15] could ameliorate renal fibrogenesis in the New Zealand obese (NZO) mouse model, which is a polygenic model of metabolic syndrome [16]. We studied the therapeutic effects of the in vivo inhibition of TRPC6 by the novel blocker SH045 in the UUO mouse model of accelerated renal fibrogenesis utilizing these mice.

## 2. Results

### 2.1. SH045 Treatment Does Not Affect Renal Function and Trpc Expression in UUO Model

To investigate the impact of in vivo TRPC6 inhibition on renal function, target molecules and fibrosis, we performed UUO in the NZO mice. During the one week period, we administrated SH045 (TRPC6 inhibitor) or vehicle once daily (Figure 1A). After 7 days, urinary tract obstruction led to hydronephrosis (Appendix A). Consistent with our previous findings, *Trpc6* expression significantly increased in UUO kidneys. SH045 affected neither *Trpc6* mRNA expression nor the expression of other TRPC channels, including *Trpc1, Trpc2, Trpc3 and Trpc4* (Figure 1B and Appendix A). SH045 had no impact on renal function. Serum creatinine (*p* = 0.1098; Figure 1B) and blood urea nitrogen (BUN) (*p* = 0.928; Figure 1C), serum cystatin C, urine albumin, and urine albumin-to-creatinine ratio in SH045-treated mice were unchanged (Figure 1D–F). In addition, we found no differences in serum levels of glucose, sodium, potassium, ionized calcium, total CO_2_, hemoglobin, hematocrit, and anion gap in SH045-treated animals compared to the vehicle group (Appendix A). SH045-treated mice exhibited a slightly higher serum chloride concentration (*p* = 0.047), albeit within the normal physiological range.

### 2.2. SH045 Treatment Does Not Alter Kidney Parenchymal Damage

Morphologically, UUO increased mesangial matrix deposition, leading to glomerular hypertrophy, and tubular dilatation (Figure 2A–D). The expressions of renal damage markers, kidney injury molecule-1 (*Havcr1*) and Lipocalin-2 (*Lcn2*), were increased in UUO kidneys compared to control (Figure 2E,F). However, SH045 did not affect these parameters in both UUO and control kidneys (Figure 2A–F). These results indicate that TRPC6 inhibition per se has no impact on the damage to renal parenchyma (glomerular or tubular) caused by UUO.

### 2.3. SH045 Treatment Ameliorates Renal Expression of Inflammatory Markers

Next, we measured the renal mRNA expression of inflammatory cytokines and chemokines using qRT-PCR. The expression of inflammatory molecules was markedly increased in kidneys subjected to UUO compared to control groups (Figure 3). The mRNA expression of chemokine (C-X-C motif) ligand 1 (*Cxcl1*), chemokine (C-C motif) ligand 5 (*Ccl5*), and chemokine (C-C motif) receptor 2 (*Ccr2*) was significantly lower in UUO kidneys of SH045-treated mice (SH045 UUO kidneys) compared to UUO kidneys of vehicle-treated mice (vehicle UUO kidneys) (Figure 3A–C). The expressions of chemokine (C-C motif) ligand 2 (*Ccl2*), chemokine (C-X-C motif) ligand 2 (*Cxcl2*), and intercellular adhesion molecule 1 (*Icam1*) were increased in both SH045 UUO and vehicle UUO kidneys compared to control kidneys, although there were no differences between SH045 UUO and vehicle UUO kidneys (*p* = 0.056, *p* = 0.068 and *p* = 0.076, respectively) (Figure 3D–F). Furthermore, immunofluorescence staining of ICAM-1 markedly increased in UUO kidneys in comparison to control kidneys (Appendix A). Additionally, the pharmacological inhibition of TRPC6 by SH045 decreased ICAM-1 expression after UUO in comparison to vehicle-treated kidneys (Appendix A). Whereas ICAM1 expression was similar in the vessels of UUO kidneys, vehicle-treated kidneys had a much higher expression in SH045-treated UUO kidneys due to more ICAM-1-positive immune cell infiltration (Appendix A).

### 2.4. SH045 Treatment Leads to Less Renal Immune Cell Infiltration

To evaluate inflammatory cell infiltration in UUO kidneys, we examined macrophages and T cell presence using immunofluorescence. Kidney cross sections were immunolabelled with the macrophage marker F4/80 and T cell marker CD4 as described previously [17]. As shown in Figure 4A,B, excessive CD4-positive cells infiltration was observed in renal interstitium of UUO kidneys in comparison to control kidneys (Figure 4A,B). Similarly, the number of F4/80-positive cells in UUO kidneys was also markedly increased compared to control kidneys (Figure 4C,D). In accordance with ameliorated inflammatory cytokine and chemokine expression, SH045 treatment decreased UUO-induced macrophage and T cell infiltration (Figure 4A–D). Thus, these data suggest that TRPC6 inhibition reduces renal inflammation in the UUO model of NZO mice.

### 2.5. SH045 Treatment Reduces Renal Expression of Fibrotic Markers

Since progressive fibrosis is a typical lesion occurring after UUO [18], we examined the impact of SH045 administration on renal fibrosis. We measured the renal mRNA expression of pro-fibrotic markers, including collagen I (*Col1a2)*, collagen III (*Col3a1)*, collagen IV (*Col4a3)*, α-smooth muscle actin (*Acta2)*, connective tissue growth factor (*Ccn2*), and fibronectin (*Fn1*). All these fibrosis-associated genes were upregulated after UUO (Figure 5A–F). Notably, SH045 treatment significantly reduced *Col1a2*, *Col3a1*, *Col4a3*, *Acta2*, *Ccn2,* and *Fn1* expressions in the UUO kidney (Figure 5A–F).

To further confirm our qPCR data, Sirius red (SR) and fibronectin immunofluorescence staining was performed. Control kidneys exhibited small SR-positive (+) areas. In contrast, UUO kidneys displayed markedly increased SR^+^ areas compared to control kidneys, indicating that UUO caused considerable collagen deposition (Figure 6A,B). SH045 effectively decreased this collagen deposition (Figure 6A,B). Similarly, immunofluorescence staining revealed increased fibronectin deposition and chromogenic immunohistochemistry increased α-smooth muscle actin (α-SMA) expression in UUO kidneys in comparison to control kidneys, which were reduced by SH045 treatment (Figure 6C–F). Taken together, these data suggest that renal fibrosis and inflammatory reactions are ameliorated in response to in vivo TRPC6 inhibition by SH045.

## 3. Discussion

Renal fibrosis is the final common outcome of progressive CKD, which is often observed in metabolic syndrome [19]. To date, there are few clinical treatments that successfully target fibrosis in CKD. Thus, developing new drug treatments is the current focus. Increasing evidence indicates that TRPC6 could play a critical role in kidney fibrosis [20]. In our previous study using *Trpc6*^–/–^ mice, we found that TRPC6 deficiency ameliorated renal fibrosis and immune cellular infiltration in the UUO model [7]. However, the results were difficult to interpret due to confounding genomic and non-genomic effects of other TRPC channels, e.g., TPRC1, TRPC3, TRPC4 and TRPC5. Previous studies identified SH045 (larixyl N-methylcarbamate) as a novel, highly potent, subtype-selective inhibitor of TRPC6 channels [15]. In our previous study, we found that the in vivo inhibition of TRPC6 by SH045 had no effects on acute kidney injury (AKI) [14]. However, there are no studies on the effects of SH045 in kidney fibrosis. In the present study, we tested the hypothesis that SH045 ameliorates UUO-accelerated renal fibrosis in NZO mice.

Our results show that SH045 ameliorates fibrotic processes in UUO kidneys. Expressions of all investigated fibrosis or fibrosis-related genes were ameliorated by SH045 treatment. The histological assessment of deposited collagen and extracellular matrix protein confirmed the expression data of the genes. Of note, renal fibrosis arises after an insult, whereas resident kidney fibroblasts and cells of hematopoietic origin differentiate into myofibroblasts [21,22,23]. Myofibroblasts acquire a contractile/proliferative phenotype upon activation by profibrotic factors and become principal kidney collagen-producing cells [24]. Considerable evidence indicates that renal inflammation plays a central role in the initiation and progression of fibrosis [19]. Myofibroblasts are regulated by a variety of means, including paracrine signals derived from lymphocytes and macrophages. Critical chemokines recruiting macrophages and lymphocytes are CCL2/CCR2, CCL5, and CXCL1/2. ICAM-1 is an endothelial- and leukocyte-associated transmembrane protein in facilitating leukocyte endothelial transmigration [25]. Interestingly, our results show that SH045 inhibits the overexpression of these chemokines and the infiltration of numerous immune cells, suggesting that TRPC6 inhibition may antagonize renal fibrosis by affecting inflammatory processes. TRPC6 is expressed in a wide range of cell types, including neutrophils, lymphocytes, platelets and the endothelium, which might be a modulator of tissue susceptibility to inflammatory injuries [26,27]. Some studies suggested that TRPC6 channels may enhance chemotactic responses by increasing Ca^2+^ concentration, which promotes actin-based cytoskeleton remodeling [28,29]. Furthermore, Ca^2+^ currents within T-lymphocytes are influenced by TRPC6, which can affect the function of T-lymphocytes [30]. Novel myeloid cell subsets could be targeted to ameliorate injury or enhance repair, including an *Arg1+* monocyte subset present during injury and *Mmp12+* macrophages present during repair [31]. It is intriguing to speculate that TRPC6 inhibition might ameliorate fibrotic processes in UUO kidneys by modulating the function(s) of theses cell types.

On the other hand, TRPC6 was also reported to contribute to fibroblast transdifferentiation and healing in vivo [32]. Thus, the beneficial effects of TRPC6 inhibition seen in the UUO model might also involve fibroblasts. A TPRC6 blockade may decrease Ca^2+^ dependent activation of MEK/ERK signaling pathway [33]. Of note, this pathway was implemented in the detrimental differentiation and expansion of kidney fibroblasts [34]. The inhibition of the ERK1/2 pathway by trametinib ameliorated UUO-induced fibrosis through the mammalian target of rapamycin complex 1 (mTORC1) and its downstream targets.

In the present study, SH045 did not affect renal function parameters in 7-day-UUO mice, which is not surprising. In this short-term UUO model, the kidney function of contralateral undamaged kidney remained preserved and compensated for the loss of the obstructed kidney at the early stage [35]. We used the NZO inbred obese mouse strain, which carries susceptibility genes for diabetes and hypertension, conditions similar to metabolic syndrome and CKD in humans [36]. Our data observed in UUO induced fibrosis in NZO mice, and thus might be of importance in mimicking human CKD pathophysiology.

Renal fibrosis involves complex interactions among multiple cells and cytokine signaling pathways. Further studies of the TRPC6 modulation of renal fibrosis using single-cell RNA sequencing could help to better understand the exact mechanism(s) of action in the different cell types. Single-cell RNA sequencing enables the precise discrimination of specific cell type(s) or cell state(s) enriched in certain conditions (e.g., UUO) [31]. Thus, selecting cellular labels based on gene expression markers could represent a novel approach to determine cell type(s) or cell state(s) predominantly influenced by the inhibition of TRPC6 (by SH045) in the UUO model. Understanding the mechanisms behind TRPC6-induced fibrogenesis is essential for developing novel therapies to slow the progression of CKD.

Our study demonstrates that the in vivo administration of SH045 ameliorates immune cell infiltration and fibrosis in NZO mice subjected to UUO, which makes SH045 a promising therapeutic drug strategy in CKD treatment for metabolic syndrome.

## 4. Materials and Methods

### 4.1. Animals

Male NZO mice (*n* = 22, NZO/BomHIDife genetic background) from Max-Rubner-Laboratory, German Institute of Human Nutrition Potsdam-Rehbrücke (Nuthetal, Germany) were used. These mice had increased weight (45.90 ± 4.11g b.w) and were previously characterized [7]. Mice were held in specific-pathogen-free (SPF) condition, in a 12:12 h light–dark cycle, with free access to food and drinking water. All experimental procedures were approved by the Berlin Animal Review Board, Berlin, Germany and followed the restrictions in the Berlin State Office for Health and Social Affairs (LaGeSo) [37]. All experiments were performed in accordance with ARRIVE guidelines [38].

### 4.2. UUO Model

UUO mouse model was performed as described earlier [7]. Briefly, NZO mice were anaesthetized by isoflurane (2.2%) supplied with air flow at approximately 350 mL/min. During the surgery mice were placed on a heating pad to prevent hypothermia. Preemptive analgesia with carprofen (5–10 mg/kg b.w) was subcutaneously used. Body temperature was maintained at 37.5 °C and monitored during surgery using a temperature controller with a heating pad (TCAT–2, Physitemp Instruments, Clifton, NJ, USA). In deep anesthesia, the anterior abdominal skin was shaved. Then, a midline laparotomy was conducted via an incision of the avascular linea alba, and the left ureter was exposed from left side. The ureter was then ligated twice close to the renal pelvis using a 5–0 polyglycolic acid (PGA) suture wire (Resorba^®^, Nürnberg, Germany). The linea alba and skin were closed separately. The wound was sanitized with a silver aluminium spray (Henry Schein^®^, Berlin, Germany), and 0.5 mL of warm (37 °C) isotonic sodium chloride solution was intraperitoneally injected. Subsequently, each mouse was placed in a cage in front of an infrared (IR) lamp and monitored until they recovered consciousness. For the following two days, mice received carprofen (2.5 mg/mL) in their drinking water (1:50) with a final concentration of 0.05 mg/mL. After surgery mice had free access to drinking water and chow. Seven days after UUO surgery, mice were sacrificed by overdose of isoflurane and cervical dislocation. The blood samples were collected for further analysis and left kidneys were removed immediately. The kidneys were divided into three portions. Upper part of the kidney tissue was frozen in isopthane. Middle part of kidney was immersed in 4% phosphate–buffered saline (PBS)-buffered formalin for histological assessment. The other left tissue was snap frozen in liquid nitrogen for RNA preparation.

### 4.3. TRPC6 Inhibitor

SH045 (Larixyl-6-N-methylcarbamate) was previously described [15]. SH045 was initially dissolved in DMSO (final concentration of DMSO is 0.5%) and then in 5% Cremophor EL^®^ solution with 0.9% NaCl and used for intraperitoneal injection (i.p.). Mice subjected to UUO were treated with SH045 (20 mg/kg once per day, i.p.) or vehicle daily until day 7 after surgery.

### 4.4. Blood Measurements and Drugs

The blood measurements of sodium, potassium, chloride, ionized calcium, total carbon dioxide, glucose, urea nitrogen, creatinine, hematocrit, hemoglobin, and anion gap were performed at endpoint. Nighty-five microliters of blood were taken from the facial vein, and parameters were measured using i-STAT system with Chem8+ cartridges (Abbott GmbH, Wiesbaden, Germany).

### 4.5. Quantitative Real-Time (qRT)-PCR

The qRT-PCR was performed as previously described [7]. Briefly, total mRNA from mice was isolated from snap-frozen kidneys using RNeasy RNA isolation kit (Qiagen, Australia), according to the manufacturer’s instructions. The concentration and quality of RNA were determined by NanoDrop-1000 spectrophotometer (Thermo Fisher Scientific, Waltham, MA, USA). Next, RNA was transcribed to cDNA using a reaction kit (Applied Biosystems, Waltham, MA, USA). Quantitative analysis of target marker was performed with qRT-PCR using the relative standard curve method. TaqMan or SYBR green analysis was conducted by using an Applied Biosystems 7500 Sequence Detector (Applied Biosystems, Waltham, MA, USA). The expression levels were normalized to 18S rRNA. All primer sequences are provided in Appendix A.

### 4.6. Kidney Histopathology

Histological kidney assessment was performed as previously reported [39]. Formalin-fixed, paraffin-embedded sections (2 μm) of kidneys were subjected to periodic acid–Schiff (PAS) and Sirius red (SR) staining. The PAS reaction visualized the basement membranes of the capillary loops of the glomeruli, through which the glomerular damage can be evaluated [7]. In each group, 10 fields of view were randomly selected from each kidney sample section under a 400× magnification, and the average ratio of glomerular section area to total area within the view was calculated using the software ImageJ. SR staining allows for a quantification of interstitial fibrosis. The severity of tubule interstitial fibrosis was graded from 0 to 3 according to the distribution of lesions: 0, no lesion; 1, less than 20%; 2, 20–50%; 3, more than 50% [40]. Semi-quantitative glomerular damage and renal fibrotic scoring were performed in a blinded manner at 400× magnification per sample. All measurements were repeated three times.

### 4.7. Immunofluorescence and Immunohistochemistry

We performed immunostaining as previously described [7,41]. Immunofluorescence or immunohistochemistry was performed on 3-µm ice-cold acetone-fixed cryosections of kidneys using the following primary antibodies: anti-fibronectin, anti-CD4, anti-F4/80, anti-ICAM-1, anti-α-SMA (AbD Serotec, Oxford, UK). For indirect immunostaining, non-specific binding sites were blocked with 10% normal donkey serum for 30 min. Then, sections were incubated with the primary antibody for 1 h at room temperature or overnight at 4 ℃. All incubations were performed in a humid chamber. For fluorescence visualization of bound primary antibodies, sections were further incubated with Cy3-conjugated secondary antibodies (Jackson Immuno Research, WG, USA) for 1 h in a humid chamber at room temperature. Slides were analyzed using a Zeiss Axioplan-2 imaging microscope with the computer program AxioVision 4.8 (Zeiss, Jena, Germany). For immunohistochemistry, after incubation with the primary antibody directed against α-SMA, biotinylated secondary antibody (Dako REAL™ EnVision™; Dako Denmark A/S, Glostrup, Denmark) was used. Immunohistochemical positive staining was consecutively revealed by the 3,3′-Diaminobenzidine Peroxidase Substrate Kit (Dako REAL™ EnVision™; Dako Denmark A/S, Glostrup, Denmark) in accordance with the manufacturer’s instructions.

Quantitative analyses of infiltrating cells (CD4+ and F4/80+) and fibroblasts (α-SMA+) were counted in 15 non-overlapping, randomly chosen fields per kidney section under a 400× magnification. The average ratio of the fibronectin or ICAM-1-labeled area to the total area in the view (400×) was calculated using the software ImagJ (NIH, Bethesda, MD, USA). In addition, ICAM-1 expression was also analyzed using software ImagJ to calculate the mean gray value (integrated density to area).

### 4.8. Statistics

Statistical analysis was performed using GraphPad 5.04 software. Study groups were analyzed by two-way ANOVA using Sidak’s multiple comparisons post hoc test. Data are presented as mean ± SD. *p* values < 0.05 were considered statistically significant.

## Figures and Tables

**Figure 1 ijms-23-06870-f001:**
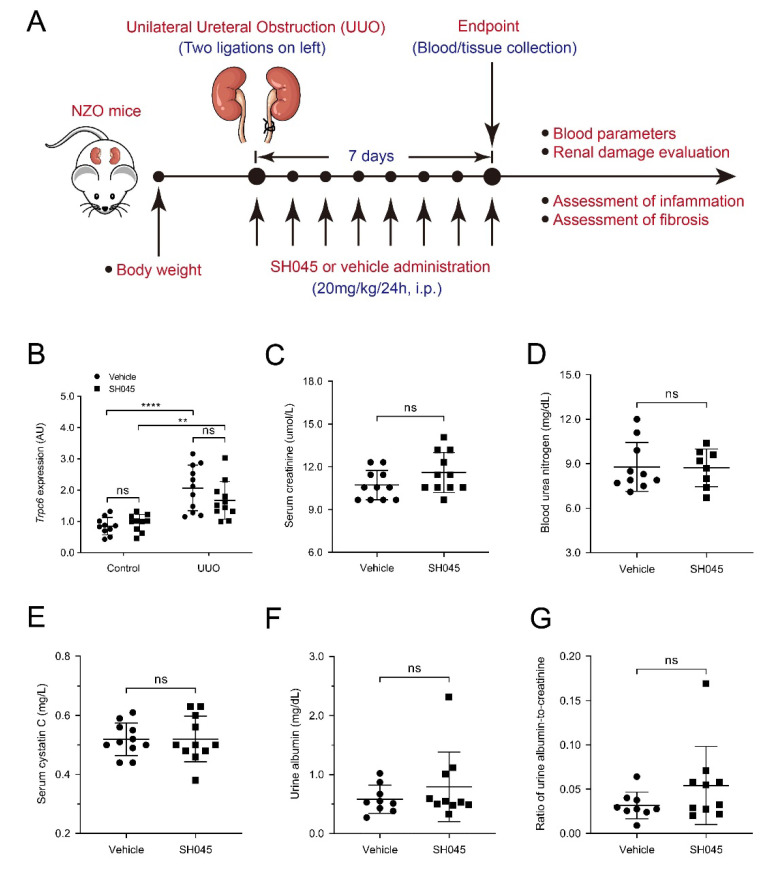
Impact of SH045 administration on renal function and Trpc6 expression in UUO model. (**A**) Experimental design of unilateral ureteral obstruction (UUO) model. NZO mice were subjected to UUO and then injected with SH045 (*n* = 11) or vehicle (*n* = 11) once every 24 h between day 0 and day 7. All mice were euthanized on day 7 after UUO surgery. (**B**) Renal mRNA levels of *Trpc6* (control *n* = 10, UUO *n* = 11). Control group includes kidneys that were not subjected to the UUO. (**C**) Serum levels of creatinine, (**D**) blood urea nitrogen, and (**E**) cystatin C in the experimental UUO groups. (**F**) Urine albumin and (**G**) ratio of albumin to creatinine in the experimental UUO groups (UUO vehicle *n* = 10–11, UUO SH045 *n* = 8–11). Data expressed as means  ±  SD. Two-way ANOVA followed by Sidak’s multiple comparisons post hoc test. ** *p* < 0.01 and **** *p* < 0.0001 defined as significant. ns, not statistically significant. AU, arbitrary units.

**Figure 2 ijms-23-06870-f002:**
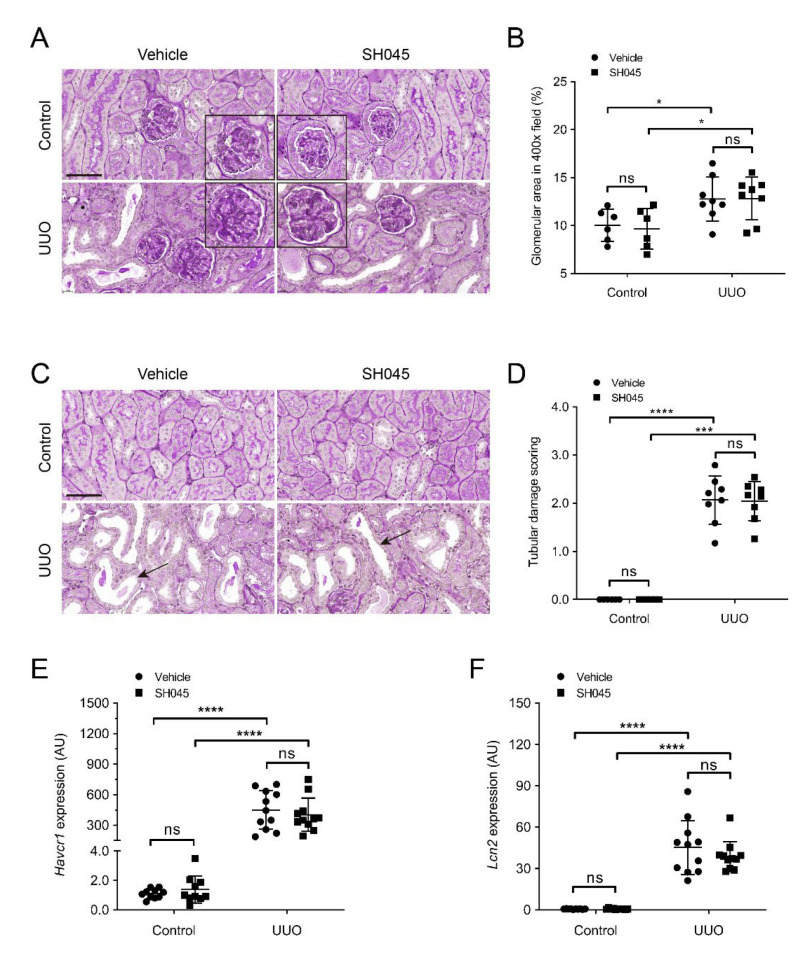
SH045 impact on kidney histopathology after UUO. (**A**) Representative images of UUO-injured glomerulus (magnification: 400×). Kidney sections were stained with periodic acid–Schiff staining (PAS). (**B**) Quantification of glomerular damage (control *n* = 6, UUO *n* = 8). (**C**) Representative images of UUO-injured tubules (magnification: 400×). Kidneys sections were stained with periodic acid–Schiff staining (PAS). Arrows indicate tubular injury. Scale bars are 50 µm. (**D**) Semi-quantification of tubular damage (control *n* = 6, UUO *n* = 8). (**E**) Renal mRNA levels of kidney injury molecule 1 (*Havcr1*) and (**F**) Lipocalin 2 (*Lcn2*) (control *n* = 10, UUO *n* = 11). Data expressed as means ±  SD. Two-way ANOVA followed by Sidak’s multiple comparisons post hoc test. * *p* < 0.05, *** *p* < 0.001 and **** *p* < 0.0001 defined as significant. ns, not statistically significant. AU, arbitrary units.

**Figure 3 ijms-23-06870-f003:**
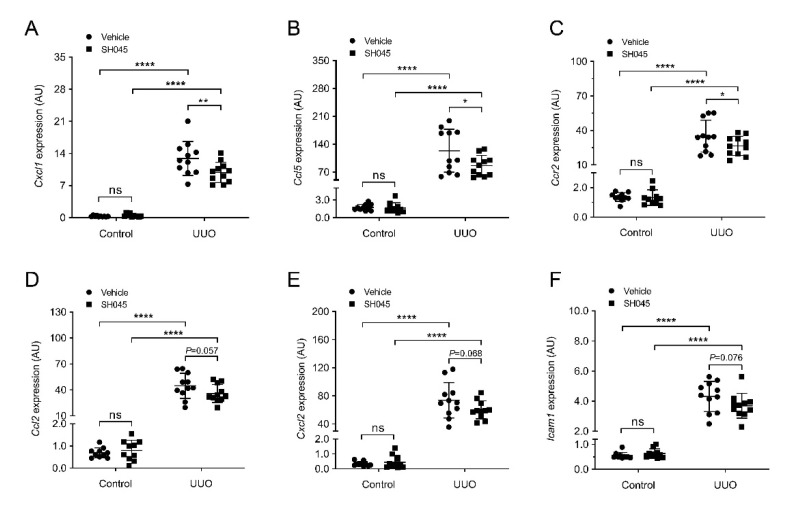
SH045 impact on renal expression of inflammatory markers. (**A**) Renal mRNA levels of chemokine (C-X-C motif) ligand 1 (*Cxcl1*), (**B**) chemokine (C-C motif) ligand 5 (*Ccl5*), (**C**) chemokine (C-C motif) receptor 2 (*Ccr2*), (**D**) chemokine (C-C motif) ligand 2 (*Ccl2*), (**E**) chemokine (C-X-C motif) ligand 2 (*Cxcl2*), and (**F**) intercellular adhesion molecule-1 (*Icam1*) (control *n* = 10, UUO *n* = 11). Data expressed as means  ±  SD. Two-way ANOVA followed by Sidak’s multiple comparisons post hoc test. * *p* < 0.05, ** *p* < 0.01, and **** *p* < 0.0001 defined as significant. ns, not statistically significant. AU, arbitrary units.

**Figure 4 ijms-23-06870-f004:**
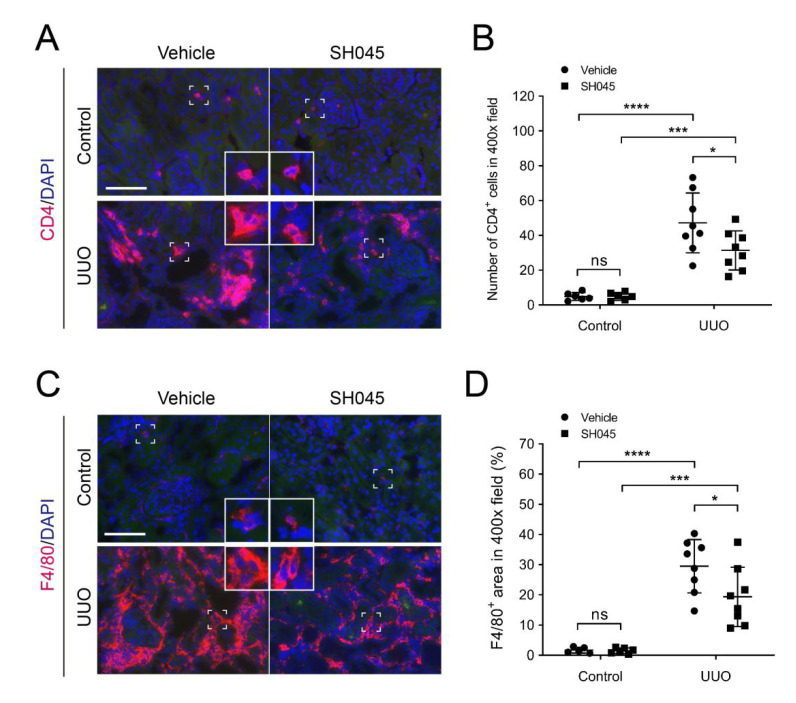
SH045 impact on renal inflammatory cell accumulation after UUO. (**A**) Representative images of control and UUO-injured kidneys stained with CD4+ T cells (magnification: 400×). Rectangles represent single-cell magnifications. Scale bars are 50 µm. (**B**) Quantification in renal infiltration of CD4+ T cells (control *n* = 6, UUO *n* = 8). (**C**) Representative images of control and UUO-injured kidneys stained with F4/80+ macrophages (magnification: 400×). Rectangles represent single-cell magnifications. Scale bars are 50 µm. (**D**) Quantification in renal infiltration of F4/80+ macrophages (control *n* = 6, UUO *n* = 8). Data expressed as means  ±  SD. Two-way ANOVA followed by Sidak’s multiple comparisons post hoc test. * *p* < 0.05, *** *p* < 0.001, and **** *p* < 0.0001 defined as significant. ns, not statistically significant. AU, arbitrary units.

**Figure 5 ijms-23-06870-f005:**
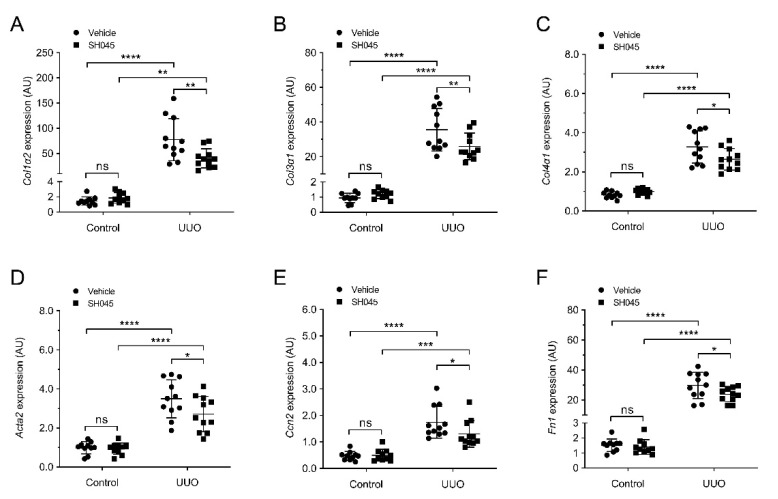
SH045 impact on expression of renal fibrotic markers UUO. (**A**) Renal mRNA levels of collagen type I α 1 (*Col1α2*), (**B**) Collagen type III α 1 (*Col3α1*), (**C**) Collagen type IV α 1 (*Col4α1*), (**D**) α-Smooth muscle actin (*Acta2*), (**E**) Connective tissue growth factor (*Ccn2*), and (**F**) Fibronectin (*Fn1*) (Control *n* = 10, UUO *n* = 11). Data expressed as means  ±  SD. Two-way ANOVA followed by Sidak’s multiple comparisons post hoc test. * *p* < 0.05, ** *p* < 0.01, *** *p* < 0.001 and **** *p* < 0.0001 defined as significant. ns, not statistically significant. AU, arbitrary units.

**Figure 6 ijms-23-06870-f006:**
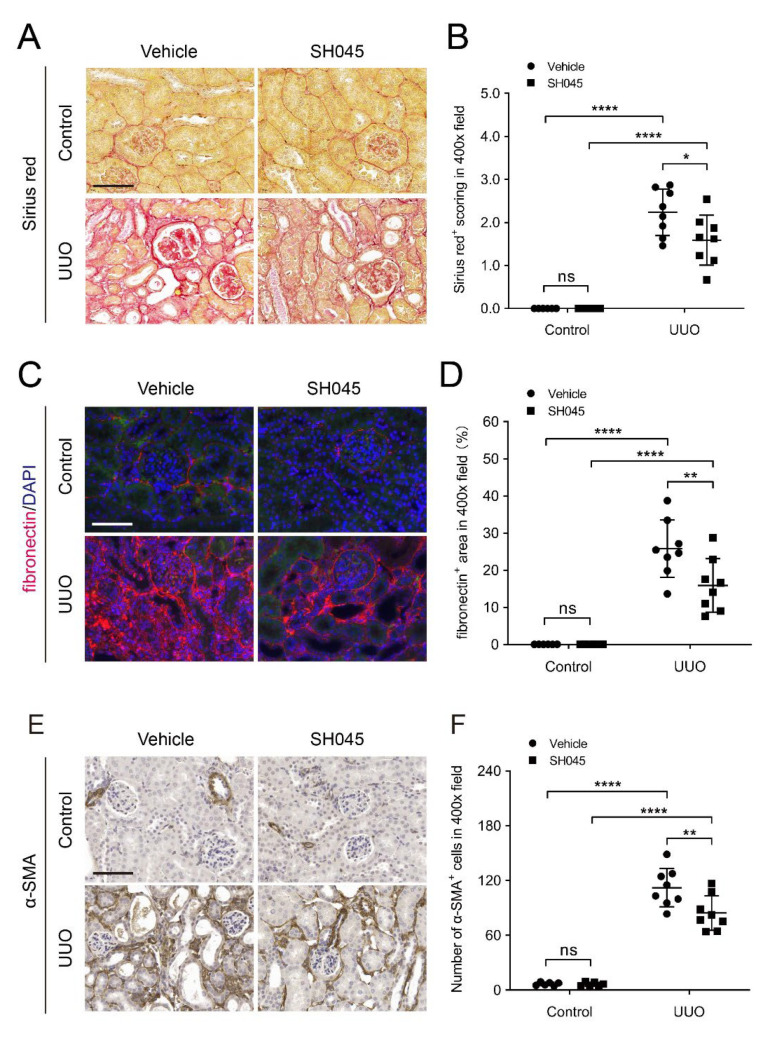
SH045 impact on renal fibrogenesis after UUO. (**A**) Representative images of control and UUO-injured kidneys stained with Sirius red (magnification: 400×). Scale bars are 50 µm. (**B**) Semi-quantification in renal Sirius red+ area proportion (control *n* = 6, UUO *n* = 8). (**C**) Representative images of control and UUO-injured kidneys stained with fibronectin (magnification: 400×). Scale bars are 50 µm. (**D**) Quantification in fibronectin+ area (control *n* = 6, UUO *n* = 8). (**E**) Representative images of control and UUO-injured kidneys stained with α-SMA (magnification: 400×). Scale bars are 50 µm. (**F**) Quantification of α-SMA^+^ staining (control *n* = 6, UUO *n* = 8). Data expressed as means  ±  SD. Two-way ANOVA followed by Sidak’s multiple comparisons post hoc test. * *p* < 0.05, ** *p* < 0.01, and **** *p* < 0.0001 defined as significant. ns, not statistically significant.

## Data Availability

Represented data are publicly archived datasets. For further information, please contact the corresponding author.

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
