# Peer review of "In Vivo Inhibition of TRPC6 by SH045 Attenuates Renal Fibrosis in a New Zealand Obese (NZO) Mouse Model of Metabolic Syndrome"

_ijms, 2022, doi:10.3390/ijms23126870_

Round 1
Reviewer 1 Report
Dr. Zheng and his/her colleagues are trying to determine whether SH045, one of an inhibitor of TRPC6 can attenuate renal fibrosis in the NZO mouse model of metabolic syndrome. The experiments were designed well and generated some great data. However, some important issues have to be addressed.
- In figure 1B, in the control or UUO model, SH045 did not impact TRPC6 expression. As an inhibitor, could you explain why SH045 did not inhibit its expression?
- SH045 treatment did not alter renal histopathology. Did the dose of SH045 is tested or not? Otherwise, it is difficult to convince your results.
- PCR is not enough to confirm all of your results. The authors have to consider using more methods from different levels to confirm their results.
- What kind of target cells will you focus on? The authors have to how SH045 affects the cells in the kidney. Otherwise, how can you use a single-cell sequence to determine the mechanisms?
- The authors have to check the mechanism.
Reviewer 2 Report
The Authors report their results about a novel experimental therapeutic option able to decrease interstitial inflammation secondary to a persistent obstructive insult in a dis metabolic kidney. The paper is well described and interesting for clinical implications.
Few thoughts and minor revisions:
1.Figure 2. The semiquantitative evaluation is quite impressive in a 5% variation grading scale especially in the glomerulus.
2. Figure1. IRI stands for? please clarify
3. Page 2, line 84. Figure 1C shows creatinine, not trpc expression. Please, correct
Reviewer 3 Report
[IJMS] Manuscript ID: ijms-1663959 - Review Request
I have read with interest the manuscript entitled “In vivo inhibition of TRPC6 by SH045 attenuates renal fibrosis 2 in the New Zealand obese (NZO) mouse model of metabolic 3 syndrome” by Zhihuang Zheng and al.
Members of the TRPC family, including TRPC6 channels, are shown as key contributors in the pathogenesis of renal and cardiovascular diseases (Ma et al 2016, Ilatovskaya et al 2015, Lin et al 2019). Previous articles, including some of the authors, have shown that TRPC6 is an important mediator of fibrosis, contributing to renal disease. In this paper, Zhihuang Zheng and collaborators further examine whether inhibition of TRPC6 with a novel selective TRPC6 inhibitor SH045 (larixyl N-methylcarbamate) impacts on renal fibrosis and inflammatory cell infiltration in an early CKD model of unilateral ureter obstruction (UUO) in a NZO mouse model. They demonstrate that SH045 exerts a renoprotective role, alleviating renal fibrosis and infiltration in their UUO mouse model. Thus, despite the lack of effects on renal function, SH045 prevents tubulointerstitial fibrosis in UUO as characterized by reduced collagen and fibronectin deposition. In addition, SH045 treatment results in decreased expression of inflammatory molecules and subsequent immune cell infiltration (macrophage and T cell infiltration) in UUO kidneys of NZO mice.
The data presented are interesting. The experimental design is well conceived and pertinently discussed. However, i have major concerns regarding the study design and the conclusions drawn from the experiments. Some issues need to be clarified in order to strengthen the study and make more convincing the results.
P1 : SH045 treatment leads to less renal immune cell infiltration and prevented tubulointerstitial fibrosis in UUO as characterized by reduced ECM deposition. Surprisingly, SH045 does not restore any kidney function parameters in NZO mice. The UUO model generates interstitial fibrosis but no glomerular disease or proteinuria. Using an NZO mouse model, that engages these abnormalities, should be a good model to test the therapeutic potential of SH045. Various studies have highlighted the important role of TRPC6 in the pathogenesis of glomerular injury using TRPC6 knock-out animal models. In most studies, TRPC6 knock-out animals showed markedly reduced glomerulosclerosis and podocyte foot effacement (Li et al 2017, Kim et al 2018, Spires et al 2018). In this context, Spires’article showed that TRPC6 knockout protect podocytes and kidney function in the streptozotocin (STZ)-treated Dahl SS rat (an established model of type 1 diabetes). Is the absence of restoration of renal function by SHO45 expected or surprising in NZO mouse model ? Has this lack of restoration of kidney function ever been observed in TRPC6 knockout mice (previous paper of the authors, Kong et al 2019)? This crucial point is not argued and explained in the discussion. Based on their results, the authors state that TRPC6 inhibition by SH045 might be a promising antifibrotic therapeutic agent whereas SH045 treatment does not improve renal histopathology or renal function. The conclusion regarding the importance of treatment with TRPC6 inhibitors should be more nuanced.
P2 : The two Controls (Vehiculed and SH045-treated contralateral non-obstructed kidneys) are missing in fig 1 (C-G) (Page 3 figure 1). Therefore, it is not possible to assess the real effect of UUO on the loss of renal function in NZO mice.
P3 : Authors demonstrate that SH045 treatment decreases ECM deposition (collagens and fibronectin) in UUO kidneys of NZO mice. This result was confirmed at mRNA and protein levels using several techniques (qPCR data, sirius red and immunofluorescence stainings). This decrease in ECM deposition by SH045 was only associated with a fall in the expression of the fibrotic marker α-smooth muscle actin (a-SMA), at mRNA levels by qPCR (Figure 5). This last result is not sufficient to conclude on a concrete decrease in the pool of myofibroblasts that represent the principal kidney collagen-producing cells. It is unclear whether SH045 treatment can reduce the number of myofibroblasts in UUO kidneys of NZO mice or have only an action on myofibroblasts activation. It would be interesting to evaluate by IHC the expression of mesenchymal markers α-SMA, S100A4/FSP-1 in the kidney cortex, which reflects myofibroblast number. In this context, a recent paper demonstrated that an orally bioavailable specific TRPC6 inhibitor, BI 749327, reduces renal fibrosis and associated gene expression (in particular αSMA and S100A4) in mice with UUO (Lin et al 2019).
P4 : No action on renal function being observed by SH045, it would be interesting to improve the manuscript to better understand how the inhibitor acts on the myofibroblastic component. Myofibroblasts could derive from renal epithelial/endothelial cells, interstitial fibroblastic cells, or mesenchymal pericytes and their relative contributions may vary depending on the type of injury and/or the animal model. Increasing evidences suggests that TRPC6 is involved in several of these mechanisms of myofibroblast production. TRPC6 is an important determinant for TGF-β1-induced myofibroblast differentiation during pulmonary fibrosis (Hofmann et al 2017). Results concerning the involvement of EMT in renal fibrosis are contradictory, attributing an important or negligible role to tubular EMT in the generation of myofibroblasts, whose intervention is however limited to diabetic nephropathy (Loeffler et al 2015). Thus, EMT process could be also involved in TIF in the NZO mouse model. It would be interesting to evaluate (by IHC, immunofluorescence analysis) the expression of epithelial (i.e Ecadherin, Cadh16, …) and mesenchymal markers (N-cadherin, vimentin …) in fibrotic kidneys from both vehicle- and SH045-treated animals. Working on Primary Culture of Kidney TEC and/or renal myofibroblasts (derived from explants of vehicle-treated UUO-kidneys) could also be important.
P5 : Therapeutic efficacy of a selective pharmacological TRPC6 inhibition of CKD was recently demonstrated in preclinical animal models using BI 749327 that is described as the most selective antagonist to date. In this article, Lin et al reported that BI 749327 prevents tubulointerstitial fibrosis in UUO in a dose-dependent manner (Lin et al 2019). We cannot exclude that the lack of renal function restoration by SH045 in NZO mice is due to the use of a single and lower dose of inhibitor (20 mg/kg once daily, i.p.). Have you tried other doses? As in the author's previous article with TRPC6 knockout mice (Kong et al 2019), it would have been informative to test the action of another TRPC6 blocker, such as BI 749327, in order to confirm the lack of recovery of renal function with both inhibitors in the NZO mouse model.
Minor comments:
Page 8 line 191 “(C) Representative images of UUO-injured kidneys stained with for?? fibronectin (magni-191 fication: 400×)”.
Page 9 lines 219-220 : “Critical chemokines recruiting macrophages and lymphocytes are CCL2/CCR2, CCL5, CXCL1/2 and ICAM-1”. ICAM-1 is not a chemokine. Please modify the sentence.
Page 9 lines 221-222 : “Interestingly, our results show that SH045 inhibits overexpression of these chemokines and infiltration of numerous immune cells, suggesting that TRPC6 may antagonize renal fibrosis by affecting inflammatory processes”. TRCP6 does not antagonize renal fibrosis but contributes to renal fibrosis.
Page 9 lines 237-238 : “Of note, this pathway 237 has been implemented in the detrimental differentiation and expansion of kidney fibro-238 blasts [33]. In this study, inhibition of the ERK1/2 pathway….” “In this study” is confusing for the readers.
Round 2
Reviewer 1 Report
- Could you show some data to confirm the changes in the ion channel?
- I appreciate your answer.
- Although you used several methods in the whole manuscript, in one figure, only PCR is not enough to confirm your conclusions.
- In the kidney, the majority of cells are not CD4+ and F4/80+ cells as shown in your control, how can you confirm the mRNA changes from CD4+ and F4/80+ cells ? Your IF results show that majority of cells are CD4+ and F4/80+ cells in UUO model. Could you show the nuclears clearly?
- The mechanism is important for your study.
Reviewer 3 Report
- I agree that there is no need to further discuss the UUO model, a well-established model of accelerated fibrosis. However, since you showed that SH045 did not affect renal function parameters (results shown in fig 1 and not in Supplementary Data), these results should be explained in the discussion so that the majority of readers who are not necessarily familiar with the UUO model, can understand the results presented in fig1.
- I am convinced that the SH045 inhibitor is effective in reducing renal fibrosis in NZO mice. The results showing that SH045 treatment decreases ECM deposition and immune cell infiltration in UUO kidneys are compelling.
However, the results of qPCR to better understand the mechanisms of action of the inhibitor on the (myo)fibroblastic component remain insufficient (Figure 5). I agree that IHC using α-SMA, S100A4/FSP-1 mesenchymal markers in renal cortex would not give a definitive answer regarding cell type and origin. But this kind of experiment (or others), will first confirm the results of qPCR using another technique and may begin to give the beginning of an answer. In my opinion, it is important to better understand in this article the mechanisms of action of the inhibitor to know if the SH045 treatment can reduce the number of myofibroblasts in the UUO kidneys of NZO mice or ONLY have an action on the activation myofibroblasts (Points 3 et 4 of the previous review report)

Round 3
Reviewer 3 Report
I think that this manuscript now fulfills the qualitative criteria requested by “International Journal of Molecular Sciences”
1/ The authors followed the recommendations allowing the readers to better understand the work, in particular concerning the IHC a-SMA.
2/ There are still small errors in the text (especially in the added text) to be corrected :
- Concerning the new sentence “Additionally, pharmacological inhibition 136 of TRPC6 by SH045 ameliorated ICAM-1 expression after UUO in comparison to vehicle-137 treated kidney (Figure S3A-C)”. lines 136-138
Vehicle-treated kidney (Figure S3A-C) line 138, it lacks an S to kidneys
The verb “to ameliorate” is very suggestive: it can say to increase or to decrease depending on the context and the molecule studied. Please replace it.
- Concerning the new sentence “Similarly, the immunofluorescence staining revealed increased fibronectin deposition and α-smooth muscle actin (α- SMA) expression in UUO kidneys in comparison to control kidneys, which were reduced by SH045 treatment (Figure 6 C-F)”. lines 190-192 : fibronectin is an immunofluorescence, α- SMA is not. Thanks for rectifying
- Figure 6 (F) Quantification of α-SMA+ staiining (Control 203 n=6 each, UUO n=8 each, respectively). Lines203-204 : 2 i to the word staining
- In several figure legends, it is specified: (Control n=6 each, UUO n=8 each, respectively). I don't understand “each” and “respectively”. We could simply mark (Control n=6, UUO n=8).
